# Dynamical Design and Gain Performance Analysis of a 3-DoF Micro-Gyro with an Anchored Leverage Mechanism

**DOI:** 10.3390/mi13081201

**Published:** 2022-07-28

**Authors:** Kunpeng Zhang, Sai Wang, Shuying Hao, Qichang Zhang, Jingjing Feng

**Affiliations:** 1Tianjin Key Laboratory for Advanced Mechatronic System Design and Intelligent Control, School of Mechanical Engineering, Tianjin University of Technology, Tianjin 300384, China; jjfeng@tju.edu.cn; 2National Demonstration Center for Experimental Mechanical and Electrical Engineering Education, Tianjin University of Technology, Tianjin 300384, China; 3Tianjin Key Laboratory of Nonlinear Dynamics and Control, School of Mechanical Engineering, Tianjin University, Tianjin 300072, China; qzhang@tju.edu.cn

**Keywords:** leverage amplification principle, gain, complete 2-DoF system, micro-gyro, nonlinearity

## Abstract

In this paper, we apply the leverage amplification principle to improve the gain of a three-degrees-of-freedom (3-DoF) micro-gyro. The gain of the micro-gyro can be improved by designing linear and nonlinear micro-gyros with an anchored lever mechanism (ALM). First, the sensor system of the micro-gyro is designed as a complete 2-DOF system with an ALM. The effect of the leverage rate (LR) on the mass ratio and frequency coupling parameter (FCP) of the complete 2-DOF sense system is studied. We analyze the variation rule of the gain of the lever’s input and output as the LR increases. Afterwards, the bandwidth and gain performance of linear and nonlinear micro-gyros with an ALM is investigated by applying the arbitrarily tunable characteristics of peak spacing of the complete 2-DOF system. The influence of LR, FCP, nonlinear strength, damping, and peak spacing on bandwidth and gain of the 3-DOF micro-gyro is analyzed. The results indicate that both LR and FCP have a large effect on the gain and bandwidth of a micro-gyro with an ALM. The LR parameter mainly improves the gain of the micro-gyro, and the FCP parameter mainly adjusts the bandwidth performance. Adding levers can effectively improve the gain performance of the linear micro-gyro. The linear micro-gyro with an ALM can improve the gain by 4.5 dB compared to the one without an ALM. The nonlinear micro-gyro with an ALM combines two characteristics: the nonlinear micro-gyro can improve the bandwidth, while the lever structure can improve the gain. Compared with the linear micro-gyro without an ALM, the gain can be increased by 17.6 dB, and the bandwidth can be improved as well. In addition, the bandwidth of a micro-gyro with an ALM is related to the gain difference between the peaks of the lever output. The increase in the gain difference leads to a flattening of the left peak, which effectively broadens the bandwidth. For nonlinear micro-gyros with an ALM, the bandwidth can be further improved by increasing the nonlinear stiffness coefficient, and better gain and bandwidth can be obtained using a vacuum package.

## 1. Introduction

Micro-gyros are a kind of inertial sensors which are used to measure angular rate or attitude angle. They are widely applied in many fields, including automotive applications for ride stabilization and rollover detection, consumer electronic applications such as video camera stabilization, virtual reality and inertial mice for computers, robotics applications, and a wide range of military applications [1].

Micro-gyros can be classified into resonant and non-resonant micro-gyros ones depending on their gain and bandwidth performance. For resonant micro-gyros, high gain can be obtained by matching the drive and sense frequencies. Zaman et al. [2] investigated the characteristics of resonant micro-gyros with different mismatch frequencies. Their results show that frequency mismatch leads to a significant reduction in the gain of the sense mode and that resonant frequency matching can be controlled by adjusting the DC voltage, although with a narrow bandwidth. The narrow bandwidth results in poor stability, causing the micro-gyro to be greatly disturbed by external signals and making it more difficult to control the DC voltage. In order to control a resonant micro-gyro to operate in the resonant state, Fan et al. [3] proposed a novel automatic mode-matching method to reduce the frequency mismatch during the operation of the micro-gyro, allowing the measured angular velocity error to be controlled within one degree. Zhang et al. [4] proposed a scheme with a heterodyne optical phase-locked loop technique based on acousto-optic modulation for resonant fiber optic micro-gyros, meaning that the resonant frequency of the micro-gyro is locked at the resonance peak. Based on an indium phosphide (InP) wafer platform, Mdc et al. [5] demonstrated a resonator–bus–resonator anti-parity time-symmetric integrated optical micro-gyro design, which makes the micro-gyro robust against external disturbances. In order to eliminate the need for complex feedback electronics for closed-loop sensing, researchers have increased bandwidth by designing non-resonant micro-gyros (i.e., multi-DOF micro-gyros) to improve stability. Bukhari et al. [6] studied a micro-gyro with a 3-DOF drive mode and a 2-DOF sense mode. Its sense mode utilized complete 2-DOF systems with adjustable peaks spacing, ensuring high gain and improving bandwidth performance. Wang et al. [7] investigated a multi-DOF micro-gyro. Their drive mode adopted a complete 2-DOF system, while the sense mode adopted dual 2-DOF systems. Their multi-DOF micro-gyro setup was able to increase the bandwidth to more than 200 Hz.

In recent years, research on the nonlinearity of micro-gyros has attracted much attention. Hao et al. [8] explored the influence of electrostatic force nonlinearity on a 2-DOF drive mode and 2-DOF sense mode micro-gyro. The results showed that the nonlinear strength of electrostatic force was enhanced with the increase in the overlapping size of the comb teeth, which led to a softening stiffness characteristic of the micro-beam in the drive direction. A reasonable design and size of the comb teeth can improve the bandwidth and gain of the sense mode to a certain extent. Han et al. [9] discussed the effects of stiffness nonlinearity and system parameters on the dynamic characteristics, sense bandwidth, and working stability of a 2-DOF drive mode and 2-DOF sense mode micro-gyro. Their results showed that the nonlinearity led to energy transfer between the various modes of multi-DOF micro-gyros. A constant excitation frequency can obtain high response gain and ensure good robustness. Lajimi et al. [10] investigated the softening nonlinearity of vibrating beam micro-gyros. They analyzed the effects of parameters such as DC bias voltage and AC drive voltage on the system response and exploited the softening properties to improve the bandwidth and gain. Li et al. [11] established a parameter amplification model of a vibrating beam micro-gyro considering the influence of size-dependent and fringing fields. The results showed that when the parametric excitation frequency is twice the resonant frequency of the system, larger values of amplitudes in both the sense and drive directions appear, yielding a better gain. Wang et al. [12] designed a 3-DOF nonlinear micro-gyro with a complete 2-DOF system in the sense direction. As the Coriolis force effect was considered in the sense direction, the frequency response of the sense mode was coupled to the nonlinearity generated in the drive direction. When the three resonant frequencies of the micro-gyro exhibited a specific positional relationship, a good bandwidth platform and higher gain could be formed by designing stiffness nonlinearity.

The gain is related to the resolution of a micro-gyro. In a micro-gyro, low gain leads to low resolution and a weak sense signal. Multi-DOF micro-gyros enjoy improved bandwidth and good stability at the expense of gain. In recent years, scholars have applied the leverage amplification principle to increase the gain of micro-gyros and micro-resonators. For example, Peng et al. [13] analyzed a single-DOF micro-gyro with an ALM. They added levers in the sensor mechanism, and investigated the variation rule of the static stiffness and resonant frequency of the micro-gyro affected by the lever. The results showed that applying the lever effectively increased the displacement of the sense mode by about 60%. Li et al. [14] added levers in the drive mechanism of a single-DOF micro-gyro. The force in the drive direction was amplified by the lever as a means of increasing the energy transferred to the sense direction in order to improve the sensor gain and accuracy. Li et al. [15] added levers to the sensor mechanism of a tuning fork micro-gyro. They designed three sets of LRs in order to verify the feasibility of amplifying the displacement of a tuning fork micro-gyro with an ALM. Apoorva et al. [16] applied a lever mechanism to design a fully implantable hearing aid. The displacement of capacitive comb teeth was amplified by the lever, thus enhancing the capacitive sensor signal. Hong et al. [17] integrated a one-stage lever mechanism into a two-axis micro-resonant accelerometer. The levers were applied to amplify the input force in order to increase the gain of the accelerometer. Wang et al. [18] improved a MEMS piezoelectric in-plane resonant accelerometer by designing it as a centrosymmetric distribution structure. A two-stage lever mechanism was used as an amplifier to verify the mass inertial force in order to improve the gain of accelerometer measurements.

Currently, there are more designs that use the leverage amplification principle for single-DOF micro-gyros and fewer designs for multi-DOF micro-gyros. In this paper, a 3-DOF linear micro-gyro with an ALM is designed. The sense peak, gain, and bandwidth of a micro-gyro affected by LR and FCP are analyzed. We design a nonlinear micro-gyro based on a 3-DOF linear micro-gyro with an ALM. We analyze the gain and bandwidth of the nonlinear micro-gyro with ALM and the affect of parameters such as LR, nonlinearity, peak spacing, and damping coefficient. Compared with linear micro-gyros without an ALM, the linear and nonlinear micro-gyros with an ALM have significantly improved gain performance and their sensor bandwidth is effectively broadened. Moreover, the gain and bandwidth enhancement of the nonlinear micro-gyro is better.

## 2. Dynamics Analysis of a 3-DOF Micro-Gyro with an ALM

### 2.1. Classification of Levers

According to the lever principle [19], a lever can be divided into three types of amplification mechanisms. The structures of three different types of levers are shown in Figure 1. Type I levers can amplify either force or displacement, but the input and output are in opposite directions. Type II levers have the same input and output direction, but can only amplify force. In [14], a Type II lever structure was applied to amplify the drive force of a resonant micro-gyro, increasing the energy transferred from drive mode to sense mode and thereby increasing the displacement of the sensor mass, obtaining higher sensitivity performance. However, this method has limitations in the application of micro-gyros, as a greater force is required to drive the mass. Compared with the above-mentioned levers, Type III levers have a displacement amplification effect and same input and output directions of motion. For the 3-DOF micro-gyro in this paper, the main focus is to explore the gain of a micro-gyro with an ALM and without an ALM under the same drive excitation. As the same input and output motion directions are required, Type III levers are used in this study.

### 2.2. Structural Design of a 3-DOF Micro-Gyro with an ALM

Figure 2 shows a schematic diagram of the 3-DOF micro-gyro with an ALM. Here, *x* is the drive direction, *y* is the sense direction, and *z* is the external angular velocity input direction. The masses *m_b_* and *m_p_*_1_ constitute one DOF in drive mode, while the masses *m_p_*_1_ and *m_p_*_2_ constitute the first DOF in sense mode. The mass *m_s_* (Figure 2a) or masses *m_p_*_3_ and *m_p_*_4_ (Figure 2b) make up the second DOF in sense mode. The drive mode and sense mode together comprise a 3-DOF system.

Figure 2a shows a 3-DOF micro-gyro without an ALM. If mass *m_s_* is divided into two masses *m_p_*_3_ and *m_p_*_4_ (i.e., *m_s_ = m_p_*_3_
*+ m_p_*_4_) and a lever is added between *m_p_*_3_ and *m_p_*_4_, the 2-DOF system is designed as a complete 2-DOF sense system with an ALM, as shown in Figure 2b. Here, the LR *B = l/L*, *k_y_*_11_, *k_y_*_12_, *k_y_*_2_, *k_y_*_3_ are the equivalent stiffnesses of the micro-beam in the sense direction, while *c_y_*_1_, *c_y_*_2_, and *c_y_*_3_ are the damping coefficients of the micro-beam in the sense direction. Masses *m_p_*_1_ and *m_p_*_2_ are defined as sense-Ⅰ and *m_p_*_3_ and *m_p_*_4_ are defined as sense-Ⅱ, where *m_p_*_3_ is connected to the lever input and *m_p_*_4_ is connected to the lever output. Because the spring *k_y_*_11_ is relatively stiff in the drive direction, the excitation force drives the masses *m_b_* and *m_p_*_1_ to vibrate along the *x*-axis. When the input angular velocity is present along the *z*-axis, the masses vibrating along the *x*-axis generate a Coriolis force along the *y*-axis. Afterwards, the Coriolis force drives the masses of sense-I and sense-II to vibrate along the *y*-axis due to the spring *k_b_*_2_ being stiffer in the sense direction. Mass *m_p_*_1_ makes the drive and sense masses move only in their respective directions under the constraint of the spring, thus realizing a decoupled design. In this process, the gain of mass *m_p_*_3_ is amplified by the lever, which increases the gain transferred to mass *m_p_*_4_ by a factor of *B*. Figure 2c shows the physical schematic of the micro-gyro with an ALM. The main structural parameters are shown in Table 1.

According to Newton’s Second Law, the dynamic equation of the 3-DOF micro-gyro with an ALM is established as follows:

Drive direction:(1)$$m_{x}\ddot{x}+c_{b}\dot{x}+k_{b}x=F_{0}\sin\left(w_{0}t\right)$$

Sense direction:

Sense-Ⅰ
(2)$$m_{Y1}{\ddot{y}}_{1}+c_{Y1}{\dot{y}}_{1}+\left(k_{Y1}+k_{Y2}\right)y_{1}-k_{Y2}y_{2}=-2m_{Y1}\Omega_{z}\dot{x}$$

Sense-Ⅱ
(3)$$\begin{array}{l} m_{p3}{\ddot{y}}_{2}+c_{Y2}{\dot{y}}_{2}-k_{Y2}y_{1}+k_{Y2}y_{2}+f_{L11}=-2\Omega_{z}m_{p3}\dot{x}\\ m_{p4}{\ddot{y}}_{3}+c_{Y3}{\dot{y}}_{3}+k_{Y3}y_{3}=f_{L12}\\ f_{L11}=B\times f_{L12}\\ y_{3}=B\times y_{2} \end{array}$$

In the above equation, $y_{1}$, $y_{2}$ and $y_{3}$ are displacements of masses *m_p_*_2_, *m_p_*_3_, and *m_p_*_4_, respectively.

Rectifying Equation (3) yields
(4)$$\left(m_{p3}+B^{2}m_{p4}\right){\ddot{y}}_{2}+\left(c_{Y2}+B^{2}c_{Y3}\right){\dot{y}}_{2}+(k_{Y2}+B^{2}k_{Y3})y_{2}-k_{Y2}y_{1}=-2m_{p3}\Omega_{z}\dot{x}$$
where *m_x_*, *c_b_*, and *k_b_* are the mass, damping, and spring coefficients in the drive direction, *F*_0_ is the amplitude of the exciting force, $\omega_{0}$ is the excitation force frequency, and $m_{x}=m_{b}+m_{p1}$, $k_{Y3}=k_{y3}$, $m_{Y1}=m_{p1}+m_{p2}$, $k_{b}=k_{b1}+k_{b2}$, $k_{Y1}=k_{y11}+k_{y12}$, and $k_{Y2}=k_{y2}$; furthermore, $x=A_{x}\sin(\omega_{0}t-\varphi )$ is the steady-state solution of drive mode. Thus, we obtain
(5)$$A_{x}=\frac{F_{0}}{\left(k_{b}\sqrt{\left({\left(1-\frac{\omega_{0}{}^{2}}{\omega_{x}{}^{2}}\right)}^{2}+4\xi_{x}{}^{2}\frac{\omega_{0}{}^{2}}{\omega_{x}{}^{2}}\right)}\right)}$$
(6)$$\phi =\arctan\left(\frac{2\xi_{x}\omega_{0}}{w_{x}\left(1-\frac{\omega_{0}{}^{2}}{\omega_{x}{}^{2}}\right)}\right),\omega_{x}=\sqrt{\frac{k_{b}}{m_{x}}},\xi_{x}=\frac{c_{b}}{2m_{x}\omega_{x}}$$ where $A_{x}$, φ, $\omega_{x}$, and $\xi_{x}$ are the amplitude, phase, resonant frequency, and damping ratios of the drive mode, respectively.

### 2.3. Design of a Complete 2-DOF Sense System with an ALM

As the complete 2-DOF sense mode and drive mode are not coupled to each other, they can be designed independently. The structural frequency of the sense system is assumed to be
(7)$$\omega_{A}^{2}=\frac{k_{Y1}+k_{Y2}}{m_{Y1}},\omega_{B}^{2}=\frac{k_{Y2}+B^{2}\cdot k_{Y3}}{m_{p3}+B^{2}\cdot m_{p4}},\omega_{C}^{2}=\frac{k_{Y2}}{\sqrt{\left(m_{p3}+B^{2}\cdot m_{p4}\right)\cdot m_{Y1}}}$$

Equation (7) has a stiffness $k_{Y2}$ term in the numerator of structural frequencies $\omega_{A}$, $\omega_{B}$, and $\omega_{C}$, however, $\omega_{C}$ has no $k_{Y1}$ and $k_{Y3}$ terms. Thus, $\omega_{A}$ and $\omega_{B}$ can be designed independently of the coupling frequency $\omega_{C}$ [6]. The mass ratio $\mu_{1}$ of the micro-gyro with ALM is
(8)$$\mu_{1}{}^{2}=\frac{m_{p3}+B^{2}m_{p4}}{m_{y1}}$$

In Equation (8), the mass ratio is a function of LR *B*.

Substituting Equations (7) and (8) into the eigenvalue equation of Equations (2) and (4), the resonant frequency of the sense mode can be solved for
(9)$$p_{1,2}^{2}\;=\frac{1}{2}\left(\omega_{A}^{2}+\omega_{B}^{2}\mp \sqrt{{\left(\omega_{A}^{2}-\omega_{B}^{2}\right)}^{2}+{\left(2\omega_{C}^{2}\right)}^{2}}\right)$$
where $\left[M_{1}\right]=\left(\begin{array}{cc} m_{Y1} & 0\\ 0 & m_{p3}+B^{2}\cdot m_{p4} \end{array}\right)$, $\left[K_{1}\right]=\left(\begin{array}{cc} k_{Y1}+k_{Y2} & -k_{Y2}\\ -k_{Y2} & k_{Y2}+B^{2}\cdot k_{Y3} \end{array}\right)$.

It is assumed that the 
parametric frequency *ω_r_* used to design the 2-DOF system is equal to the distance between the two resonant frequencies of the sense mode.

Therefore, the resonant frequency of the sense mode is $p_{1,2}=\omega_{r}\mp \Delta /2$, where $\Delta =p_{2}-p_{1}$ and Δ is the peak spacing. Substituting these into Equation (9), we obtain
(10)$$\omega_{A,B}{}^{2}=\frac{1}{4}\left({△}^{2}+4\omega_{r}^{2}\pm 4\sqrt{-\omega_{C}^{4}+{△}^{2}\omega_{r}^{2}}\right)$$

The constraint condition for the establishment of Equation (10) is ${△}^{2}\omega_{r}^{2}-\omega_{C}^{4}\geq 0$. By introducing the FCP $\epsilon_{1}$, set $\omega_{C}^{2}=\epsilon_{1}△\omega_{r}(0\leq \epsilon_{1}\leq 1)$ and then substituting it into Equation (10), we obtain
(11)$$\omega_{A,B}{}^{2}=\omega_{r}^{2}+{\left(\frac{△}{2}\right)}^{2}\pm \omega_{r}△\sqrt{1-\epsilon_{1}{}^{2}}$$

Substituting Equations (7) and (8) into Equation (11), the stiffness of the beam in the sense direction can be found as follows:(12)$$\begin{array}{l} k_{Y2}=\epsilon_{1}\Delta \sqrt{\left(m_{p3}+B^{2}m_{p4}\right)m_{Y1}}\omega_{r};\\ k_{Y1}=m_{Y1}\omega_{A}^{2}-k_{Y2};\\ k_{Y3}=\frac{\left(m_{p3}+B^{2}m_{p4}\right)\omega_{B}^{2}-k_{Y2}}{B^{2}} \end{array}$$

Based on the known $m_{Y1,},m_{p3},m_{p4},B,\omega_{r}$, the stiffness coefficient of the sensor beam can be calculated by defining the peak spacing, Δ. The gain of the Coriolis peak has a large influence on the sense gain of a micro-gyro, and the gain of the Coriolis peak has a maximum value with respect to the FCP, $\epsilon_{1}$ [20]. In order to further investigate the effect of FCP on the Coriolis peak gain, we apply the transfer function method to solve the frequency response of Equations (2) and (4), which yields
(13)$$\frac{B1}{\Omega_{z}}=\left|-\frac{B^{2}m_{p4}\left(s^{2}+\omega_{B}^{2}\right)+m_{p3}\left(s^{2}+\omega_{B}^{2}+\omega_{C}^{2}{µ}_{1}\right)}{\nabla (\omega_{0})}\right|$$
(14)$$\frac{B2}{\Omega_{z}}=\left|-\frac{B^{2}m_{p4}\omega_{C}^{2}+m_{p3}\left(\omega_{C}^{2}+s^{2}\mu_{1}+\omega_{A}^{2}\mu_{1}\right)}{\nabla (\omega_{0}){µ}_{1}}\right|$$
where $\nabla (\omega_{0})=\left(m_{p3}+B^{2}m_{p4}\right)\left(s^{4}+s^{2}\omega_{B}^{2}+\omega_{A}^{2}\left(s^{2}+\omega_{B}^{2}\right)-\omega_{C}^{4}\right)$.

The frequency response equation of the mass *m_p_*_4_ connected to the lever output is as follows:(15)$$y_{3}=B*\frac{B2}{\Omega_{z}}=B*\left|-\frac{B^{2}m_{p4}\omega_{C}^{2}+m_{p3}\left(\omega_{C}^{2}+s^{2}\mu_{1}+\omega_{A}^{2}\mu_{1}\right)}{\nabla (\omega_{0}){µ}_{1}}\right|$$

Substituting $s=i\omega_{x}$ (where *i* is imaginary units) and Equation (11) into Equation (15), the Coriolis peak gain at the lever output of sense-Ⅱ is obtained as
(16)$$G_{2}=-\frac{4\left(4B^{2}\epsilon_{1}m_{p4}\omega_{r}+m_{p3}\left(△{µ}_{1}+4\left(\epsilon_{1}+\sqrt{1-\epsilon_{1}{}^{2}}{µ}_{1}\right)\omega_{r}\right)\right)}{△{µ}_{1}\left(m_{p3}+B^{2}m_{p4}\right)\left({△}^{2}-16\omega_{r}^{2}\right)}$$

The peak spacing Δ is set to 230 Hz, while the LR *B* values are chosen to be 1 (no amplification of gain at *B* = 1) and 4, respectively. According to Equation (16), the relationship between the gain, *G*_2_, and FCP, $\epsilon_{1}$, is shown in Figure 3.

In Figure 3, curves 1, 2, and 3 show the gain versus FCP for mass ratios equal to 0.1, 0.5, and 1, respectively. It can be seen that as the mass ratio decreases, the Coriolis peak gain increases gradually. In order to find the maximum value of the Coriolis peak at the lever output of sense-II, Equation (16) can be differentiated with respect to $\epsilon_{1}$ and made equal to zero to find the extreme point, $\epsilon_{ep}$; thus, we obtain
(17)$$\epsilon_{ep}=\frac{m_{p3}+B^{2}m_{p4}}{\sqrt{m_{p3}^{2}+\mu_{1}{}^{2}m_{p3}^{2}+2B^{2}m_{p3}m_{p4}+B^{4}m_{p4}^{2}}}$$

Equation (17) shows that the extreme point $\epsilon_{ep}$ is a function of LR. It implies that $\epsilon_{ep}$ is influenced by changes in LR.

As both the mass ratio $\mu_{1}$ and extreme point $\epsilon_{ep}$ are functions of LR *B* in a micro-gyro with an ALM, according to Equations (8) and (17) the relationship between $\mu_{1}$ and $\epsilon_{ep}$ with respect to *B* can be obtained at different masses *m_p_*_4_, as shown in Figure 4 and Figure 5, respectively. The parameters of the linear part analysis are selected as Δ=230, $\omega_{x}=\omega_{r}=5400\times 2\pi$ rad/s.

It can be seen from Figure 4 that the mass ratio shows a gradual upward trend with the increase of LR. The smaller mass *m_p_*_4_ is, the slower the increasing trend of mass ratio. This means that the micro-gyro has greater potential to improve the gain. This is due to the gain being higher when the mass ratio is smaller (Figure 3). As a result, as small a mass *m_p_*_4_ as possible should be chosen in the design. It can be seen from Figure 5 that the extreme point $\epsilon_{ep}$ of FCP is in the range of (0.92,1). With increasing LR, the $\epsilon_{ep}$ increases and approaches 1. If the selected mass *m_p_*_4_ is smaller, the trend of the extreme point $\epsilon_{ep}$ appears to be smoother. In summary, the choice of mass *m_p_*_4_ has a significant impact on the potential lever amplification of gain. In order to better improve the gain of the micro-gyro in sense mode, the mass *m_p_*_4_ should be selected in order to be as small as possible.

### 2.4. Effect of LR and FCP on the Gain of Peak

The effect of LR on the sense gain of the multi-DOF micro-gyro varies as well. Here, in order to study the effect of LR on the gain of sense-Ⅰ and sense-Ⅱ, the LR *B* is chosen to be 1.4, 2.4, and 4.4, respectively.

Based on Equations (13)–(15), the frequency response of the gain of the sense mode at different LR values is calculated as shown in Figure 6. Figure 6a shows that the Coriolis peak gain of sense-Ⅰ gradually increases as the LR increases, while the gain of the peaks on both sides gradually decreases. The gain between peaks (the gain of the valley) is relatively increased. Figure 6b shows that the overall gain at the lever input of sense-Ⅱ is reduced. The above conclusion is due to the mass ratio becoming larger with increasing LR (see Figure 4), leading to an improvement in the gain of sense-Ⅰ and a decrease in the dynamic amplification at the lever input of sense-Ⅱ. This conclusion is consistent with reference [21]. As the LR increases from 1.4 to 2.4, the gain at the lever output of sense-Ⅱ increases significantly. The Coriolis peak becomes more prominent, while the peaks on both sides gradually flatten out. However, the gain is not increased significant when LR increases from 2.4 to 4.4, as shown in Figure 6c. This is due to the fact that the gain at the lever output is equal to the gain at the lever input multiplied by LR (see Equation 15), and the gain at the lever input decreases as LR increases.

The above analysis shows that the LR has different effects on the left peak and Coriolis peak at the lever output of sense-Ⅱ. In order to analyze in more detail how the peak gain (sense gain) at the lever output of sense-II is affected by the FCP and LR, the FCP is selected as 0, 0.4, 0.8, and 1, while the LR is selected as 1.4, 2.4, and 4.4, respectively. The variation rule of the gain of each peak can be obtained as shown in Figure 7 and Figure 8.

If the excitation frequency in Equation (15) is equal to the low-order resonant frequency of sense mode, i.e., $\omega_{0}=p_{1}$, a 3D diagram of the left peak versus the LR and FCP can be obtained. Afterwards, Equation (15) can be differentiated for *B* and $\epsilon_{1}$, respectively, and the variation relationship of left peak with respect to LR and FCP can be analyzed, as shown in Figure 7. In Figure 7a,b, it can be seen that the left peak has a maximum value with respect to *B*, and the value of *B* corresponding to the maximum value increases as FCP increases. Observing Figure 7b, it can be seen that the value of *B* corresponding to the maximum value of the left peak is less than 2. However, in order to improve the gain of the sense mode of the micro-gyro, a larger LR should be selected. If $\epsilon_{1}$ is equal to 1, the left peak takes the maximum value, corresponding to the value of, *B* taken as 1.4. Thus, when *B* is greater than 1.4, the gain of the left peak decreases as *B* increases. Figure 7c shows that the derivatives of left peak with respect to $\epsilon_{1}$ are all greater than zero. This implies that the gain of the left peak increases with $\epsilon_{1}$ and grows rapidly within $\epsilon_{1}\in \left(0.92,1\right)$. Therefore, the gain of the left peak becomes lower with increasing LR and higher with increasing FCP.

In order to analyze the variation rule of the gain of the Coriolis peak with the LR and FCP, the above steps can be repeated such that the excitation frequency in Equation (15) is equal to the resonant frequency of the drive mode, i.e., $\omega_{0}=\omega_{x}$, as shown in Figure 8. In Figure 8a,b, if $\epsilon_{1}\in (0,0.92)$, the Coriolis peak first increases and then decreases with the increase of LR. If $\epsilon_{1}\in (0,0.92)$, the Coriolis peak becomes higher with increasing *B* and gradually becomes flat. Figure 8c shows that the derivative is greater than zero for *B* = 4.4. The Coriolis peak becomes higher gradually as $\epsilon_{1}$ increases, which means that the extreme point of the Coriolis peak is infinitely close to 1 at this time. When *B* is small (*B* = 1.4), the derivative decreases from greater than zero to less than zero, which means that the Coriolis peak increases first and then decreases as $\epsilon_{1}$ increases, i.e., there is an extreme point $\epsilon_{ep}$, and $\epsilon_{ep}\in \left(0.92,1\right)$. This verifies that the Coriolis peak gain becomes higher as *B* increases when $\epsilon_{1}\in (0,0.92)$.

The above analysis shows that the left peak decreases with increasing *B* and improves with increasing $\epsilon_{ep}$. The two parameters have opposite effects on the left peak of the micro-gyro. When $\epsilon_{ep}\in \left(0.92,1\right)$, the Coriolis peak increases with *B*. However, combined with Figure 6c and Figure 8, it can be seen that the Coriolis peak does not increase infinitely with *B*. Therefore, the value of *B* should be chosen reasonably. For a 3-DOF micro-gyro system, through investigating how the peak gain of the sense mode is influenced by the LR and FCP, the gain of the valley can be enhanced by adjusting the height of the peak.

## 3. Gain Performance Analysis of a Micro-Gyro with an ALM

The effect of LR and FCP on the gain of the peaks at the lever output of sense-Ⅱ in the previous section shows that the left peak becomes higher with increasing FCP, whereas the Coriolis peak becomes higher with increasing LR when $\epsilon_{1}\in (0,0.92)$. As the valley of the 3-DOF micro-gyro has good bandwidth performance, the valley is chosen as the bandwidth. It is necessary to investigate whether the valley near each peak of the micro-gyro with ALM has good gain effect. Thus, $\epsilon_{1}$ is set to $0.4,\;0.8,\epsilon_{ep}$, and 1, respectively, for analysis. This section mainly analyzes the gain and bandwidth performance of the micro-gyro with an ALM through the frequency response of the lever output of sense-Ⅱ. The bandwidth is selected using Equation $S\left(\omega \right)-S\left(\omega_{0}\right)=3\mathrm{dB}$, where $S\left(\omega \right)=20\log\left(y_{3}\right)$, $S\left(\omega \right)$ is the gain, and *y*_3_ is the displacement at the lever output.

### 3.1. Analysis of the Linear Micro-Gyro

In order to facilitate comparison with the nonlinear micro-gyro, the frequency response at the lever output of sense-Ⅱ is analyzed for a linear micro-gyro system to investigate the amplification effect of increasing LR on the gain. Here, the LR is selected as 1.4, 2.4, and 4.4 respectively. The stiffness of the support spring can be calculated using Equation (12). In accordance with [22], we select the pressure P=10 Pa. The frequency response curve can be obtained by substituting $s=i\omega_{0}$ into Equation (15), as shown in Figure 9 and Figure 10.

The gain and bandwidth data for the linear micro-gyro with an ALM are shown in Table 2. Combined with Table 2 and Figure 9, it can be seen that for the same LR, the overall gain of the sense mode is lower for smaller values of FCP ($\epsilon_{1}=0.4$). Even if a larger LR is selected (*B* = 4.4), as shown by curve 3 in Figure 9a, the overall gain is not significantly improved, and that there is no amplification effect of gain at this time. Figure 9b shows that the higher gain at the Coriolis peak leads to better amplification of the overall sense gain when FCP is set to a larger value, while the gain of the right valley increases more significantly compared to that of the left valley.

In Figure 10a, when the LR *B* = 2.4 is compared with the case of *ϵ*_1_ = 1, the gain of the left valley is higher when $\epsilon_{1}$ = $\epsilon_{ep}$($\epsilon_{ep}=0.95$). However, when the LR *B* = 4.4, the gain of the left valley is lower when $\epsilon_{1}$ = $\epsilon_{ep}$($\epsilon_{ep}=0.98$), as shown in Figure 10b. The above opposite situation occurs because the gain of the Coriolis peak at *B* = 4.4, $\epsilon_{1}$ = $\epsilon_{ep}$ is close to that at *B* = 4.4, $\epsilon_{1}$= 1, while the left peak becomes higher as $\epsilon_{1}$ increases (see Figure 7c). At this time, setting $\epsilon_{1}$ as 1 will relatively increase the gain of the left valley, and there will be a special case in which the gain of the left valley becomes lower as $\epsilon_{1}$ decreases (see *B* = 2.4 and 4.4 in Table 2). It should be noted that the data for gain and bandwidth in Table 2 are compared with the linear micro-gyro without an ALM (*B* = 1), where the gain column is the lowest gain point of 3 dB.

The third column in Table 2 shows the percentage increase of the displacement $y_{3}$ at the lever output; for example, when *B* = 2.4 and $\epsilon_{1}$= 1, the gain increases by 3.4 dB compared to without an ALM, while $y_{3}$ increases by 47.1% according to the Equation $S\left(\omega \right)=20\log\left(y_{3}\right)$.

It can be seen from Table 2 that when the FCP $\epsilon_{1}$ is set to $\epsilon_{ep}$ or 1, the linear micro-gyro with an ALM has a good gain amplification effect. For example, the gain is improved by 3.4 dB and 4.5 dB when *B* is set to 2.4 and 4.4, respectively, and the displacement of the lever output is improved by about 47–66%. Moreover, as the LR increases, the gain performance of the micro-gyro improves significantly. However, the second column in Table 2 shows that the sense gain improvement gradually becomes slower as the LR increases. For instance, when $\epsilon_{1}$ is set to 1 and *B* = 2.4 the gain is improved by 2.4 dB compared to *B* = 1.4; however, *B* = 4.4 improves the gain by only 1.4 dB compared to *B* = 2.4. In addition, the bandwidth is increased to an extent compared to the micro-gyro without an ALM, although the width of the bandwidth depends on the peak-to-peak gain difference (see Figure 10) and becomes narrower as *B* increases.

For the linear micro-gyro system with an ALM, the gain of the micro-gyro can be improved using the leverage amplification principle; the gain of the right valley improves more significantly. A larger FCP value should be selected in order to avoid negating the gain amplification effect. When $\epsilon_{1}$ is larger, the sensor gain of the micro-gyro is mainly affected by the LR, and is mainly reflected in the height of Coriolis peak. In addition, the FCP $\epsilon_{1}$ mainly adjusts the width of the bandwidth. Only when the LR is large and causes $\epsilon_{ep}$ to approach 1 does $\epsilon_{1}$ take a value greater than $\epsilon_{ep}$ to boost the gain. For example, when *B* = 4.4, $\epsilon_{1}$= 1, the gain is increased to −169.7 dB, which is optimal (as in Table 2), and its bandwidth is 94 Hz, which is only 4 Hz lower than *B* = 2.4 and $\epsilon_{1}$ = $\epsilon_{ep}$. However, compared with *B* = 4.4, $\epsilon_{1}$ = $\epsilon_{ep}$, the gain is increased by 0.2 dB and the bandwidth is increased by 4 Hz.

### 3.2. Analysis of the Nonlinear Micro-Gyro

Because the lever is added in the sense system, the nonlinear equation of the drive mode is the same as the equation of the 3-DOF nonlinear micro-gyro without an ALM in reference [12]. Therefore, the dynamic Equation (1) of the drive mode is rewritten as follows:(18)$$m_{x}\ddot{x}+c_{b}\dot{x}+k_{b}x+k_{d}x^{3}=F_{0}\sin\left(w_{0}t\right)$$
(19)$$\ddot{x}+C_{B}\dot{x}+\omega_{x}^{2}x+K_{D}x^{3}=F\sin\left(w_{0}t\right)$$
where, $\omega_{x}^{2}=\frac{k_{b}}{m_{x}},C_{B}=\frac{c_{b}}{m_{x}}=2\xi_{x}\omega^{2},\xi_{x}=\frac{c_{b}}{2m_{x}\omega_{x}},K_{D}=\frac{k_{d}}{m_{x}},F=\frac{F_{0}}{m_{x}}$.

Equation (19) is the forced 
vibration of a single-DOF damped Duffing system under harmonic excitation. By introducing 
the detuning parameter σ, the approximate periodic response of the primary 
resonance can be analyzed using the multiple timescales method [23]. Referring to 
reference [12], 
the equations solved by applying the multiple timescales method are as follows:
(20)$$16{\left(c_{B}A_{x}\omega_{x}\right)}^{2}+{\left(3k_{D}A_{x}^{3}-8\omega_{x}A_{x}\sigma \right)}^{2}=16f^{2}$$

Design of a nonlinear micro-gyro requires the interaction of the left and Coriolis peaks to produce high gain and wide bandwidth. An exploration of the gain and bandwidth of a nonlinear micro-gyro affected by the spacing between left and Coriolis peaks is shown in Figure 11. The different values of LR *B* lead to different height differences between the Coriolis peak and left peak, thus, *B* is taken as 1.4 and 4.4 for comparison, where $k_{d}=12.2$, $\epsilon_{1}=\epsilon_{ep}$.

Comparing the three 
response curves in Figure 11, it can be seen that when the distance between the 
Coriolis peak and left peak is close (curves 2 and 3), a stable high gain can 
be obtained over a wide frequency range. When the Coriolis peak is located in 
the middle of the two sense peaks (curve 1), the bandwidth range of the high 
gain is significantly reduced. When LR *B* is set to 1.4 (in Figure 11a) and 4.4 (in Figure 11b), a good bandwidth platform can be obtained by setting the resonant frequency *ω_x_* of the drive mode to 5320 × 2*π* rad/s and 5300 × 2*π* rad/s, respectively. However, comparing the two cases, it is 
found that a larger LR results in a narrower response platform bandwidth than a 
smaller LR, and results in a higher gain. Therefore, the design requirements 
for bandwidth and gain can be satisfied by selecting a suitable LR and 
frequency spacing between the Coriolis peak and left peak.

The gain and bandwidth of a micro-gyro are mainly affected by LR and FCP. The influence of *B* and $\epsilon_{1}$ on the shape of the nonlinear frequency response curve is shown in Figure 12 and Figure 13. In Figure 12 the FCP $\epsilon_{1}$ is set as $\epsilon_{ep}$, and in Figure 13 the LR *B* is set as 2.4.

Figure 12 shows that for a constant spacing between the left and Coriolis peaks, the LR and nonlinear coefficient have little effect on the shape of the frequency response curve, while the spacing between the left and Coriolis peaks plays a decisive role in the shape of the frequency response curve. Compared with Figure 12b,c, it is found that the nonlinear micro-gyro with an ALM is very sensitive to nonlinearity and that the bandwidth is significantly improved when the nonlinear coefficient is 12.5 compared to 12.2.

Figure 13 shows that the FCP has little effect on the shape of frequency response curve, and is similar to the linear case (see Figure 9), i.e., the overall gain is better when $\epsilon_{1}$ is set to a larger value.

The above analysis shows that the shape of the frequency response is only affected by the spacing between the peaks, and increasing of nonlinear coefficient can further improve the bandwidth. When the FCP is larger, the overall gain effect is better. Therefore, we set $\omega_{x}=5320\times 2\pi \;rad/s,\omega_{r}=5400\times 2\pi \;rad/s$ and Δ=230 to make the Coriolis peak close to the left peak. This makes it possible to investigate how the gain and bandwidth of the nonlinear micro-gyro with an ALM is affected by LR and FCP, as shown in Figure 14 and Figure 15.

Figure 14 shows that the gain of the nonlinear micro-gyro with an ALM gradually increases as the LR is gradually increased. When *B* is set to 1.4, although the 3 dB minimum gain of the micro-gyro is lower (see Table 2), the gain is reduced by 1.1%; there is actually a wide platform in its gain and a relative increase in overall gain. This is because the 3 dB bandwidth is selected downwards from the highest point of gain, while the left peak becomes lower as the LR increases.

As can be seen in Figure 15, the response is approximately synthesized as one peak when the Coriolis and left peaks are close to each other, i.e., there is no increase of the left peak to boost valley gain when $\epsilon_{1}$ is set to 1 (comparing to Figure 10). It is worth noting that the micro-gyro can obtain a better bandwidth platform when $\epsilon_{1}$ is set to 1. Therefore, for the nonlinear micro-gyro with an ALM the FCP $\epsilon_{1}$ is mainly used to adjust the sense bandwidth and cannot yet enhance the gain performance.

The gain and bandwidth data for the nonlinear micro-gyro with an ALM are shown in Table 3, and the data are compared to the nonlinear micro-gyro without an ALM (*B* = 1).

Combining Table 3 and Figure 14, it can be seen that the nonlinear micro-gyro with an ALM has a good gain amplification effect. Although the proportional increase of displacement $y_{3}$ at the lever output is not as high as in the linear case, the nonlinear system without an ALM inherently has higher gain than the linear micro-gyro without an ALM. For example, when *B* = 1, the gain of the linear micro-gyro is −174.2 dB and the gain of the nonlinear micro-gyro is −159.5 dB. The gain of the nonlinear micro-gyro is improved by 14.7 dB compared to the linear system, which is due to the effect produced by the small spacing between the Coriolis and left peaks. The optimal gain of the linear micro-gyro with an ALM in Table 2 is −169.7 dB, which is an improvement of only 4.5 dB compared to the linear system without an ALM. However, the optimal gain of the nonlinear micro-gyro with an ALM in Table 3 is −156.6 dB, which is 17.6 dB higher than the linear system without an ALM. The nonlinear micro-gyro with an ALM combines the characteristics of nonlinearity and the lever amplification effect to greatly increase the gain. In addition, observing the bandwidth in the fourth column in Table 3, the bandwidth of the nonlinear micro-gyro with an ALM is narrowed with the increasing LR. When *B* = 4.4 and $\epsilon_{1}=\epsilon_{ep}\left(0.98\right)$, its bandwidth is 86 Hz. Compared with other cases, although sacrificing a certain bandwidth, its gain is greatly improved to −156.6 dB.

### 3.3. Effect of Nonlinear Stiffness and Damping Coefficient on Gain and Bandwidth

The nonlinear stiffness has a certain effect on the bandwidth [24]. According to [25], the dimensions of the straight beam can be designed in order to obtain different nonlinear stiffness coefficients at large deformations. The bandwidth and gain of a micro-gyro are affected by the nonlinear stiffness, as shown in Figure 16, where $B=2.4,\epsilon_{1}=\epsilon_{ep}$.

Figure 16 shows that as the nonlinear stiffness coefficient increases, the bandwidth of the micro-gyro gradually increases while the gain at the peaks decreases slightly. This indicates that the bandwidth of a micro-gyro with an ALM can be effectively increased by enhancing the nonlinearity.

The gain and bandwidth performance of the micro-gyro have a strong relationship with the damping coefficient; in particular, the gain is limited by the damping coefficient. For frame-type micro-gyros, the damping coefficient is affected by the variation of the package pressure [12]. The package pressure may vary due to different operating environments, such as mechanical shock, vibration, and high temperature. Different package pressures were selected here in order to investigate the effect of the damping coefficient for a nonlinear micro-gyro with an ALM, as shown in Figure 17.

When the damping coefficient in Figure 17a is lower, the bandwidth of the nonlinear micro-gyro without an ALM (*B* = 1) is about 151 Hz and the bandwidth of the nonlinear micro-gyro with an ALM (*B* = 2.4) is about 155 Hz. When the damping coefficient is larger, as in Figure 17b, the bandwidth of the nonlinear micro-gyro without an ALM is 74 Hz and the bandwidth of the nonlinear micro-gyro with an ALM is 58 Hz. It is found that after increasing the damping coefficient, the bandwidth of the nonlinear micro-gyro without an ALM is reduced by 51% and the bandwidth of the nonlinear micro-gyro with an ALM is reduced by 62.6%. In contrast, as the damping coefficient of the system increases, the peak gain becomes significantly lower and the 3 dB bandwidth decreases. The bandwidth is narrower with an ALM than without an ALM. Therefore, for a multi-DOF micro-gyro with an ALM, increasing the damping coefficient of the system reduces peak gain, suppresses system nonlinearity, and reduces the bandwidth of the system. In conclusion, a vacuum package of nonlinear micro-gyros with an ALM can achieve better gain and bandwidth.

## 4. Conclusions

In this paper, the leverage amplification principle is applied to design the sensor structure of a 3-DOF micro-gyro. Taking advantage of a complete 2-DOF sense system with arbitrary adjustable peak spacing, we investigated the variation rule of the gain and bandwidth of linear and nonlinear micro-gyros with an ALM.

For linear and nonlinear micro-gyros with an ALM, the LR can effectively improve the gain of the micro-gyro. However, the mass ratio of the 2-DOF system increases with LR, resulting in a diminishing gain enhancement effect. The FCP mainly adjusts the bandwidth performance of the micro-gyro. If FCP is chosen to be larger, the bandwidth can be widened while retaining the gain amplification effect. In the linear system, as the Coriolis peak and left peak do not merge into one peak, $\epsilon_{ep}$ converges to 1 when *B* is large, and $\epsilon_{1}\;$ can be set to 1 in order to obtains the maximum effect on gain.

For linear micro-gyros, adding levers can effectively improve gain performance. Compared with a linear system without an ALM, a linear micro-gyro with an ALM can improve gain by 4.5 dB. A nonlinear micro-gyro with an ALM combines the characteristics of nonlinearity and the lever amplification effect. Compared with the linear system without an ALM, gain is improved by 17.6 dB, and it can greatly enhance the gain effect of a nonlinear micro-gyro.

In addition, the bandwidth of a micro-gyro with an ALM is related to the gain difference between peaks at the lever output of sense-Ⅱ. Increasing the gain difference leads to a flattening of the left peak thus widening the bandwidth. For a nonlinear micro-gyro with an ALM, increasing the nonlinear coefficient *k_d_* can further improve bandwidth performance. Therefore, the leverage amplification principle leads to a multi-DOF micro-gyro with high gain performance while effectively widening the sensor bandwidth. In addition, a vacuum package of nonlinear micro-gyros with an ALM can achieve better gain and bandwidth.

## Figures and Tables

**Figure 1 micromachines-13-01201-f001:**
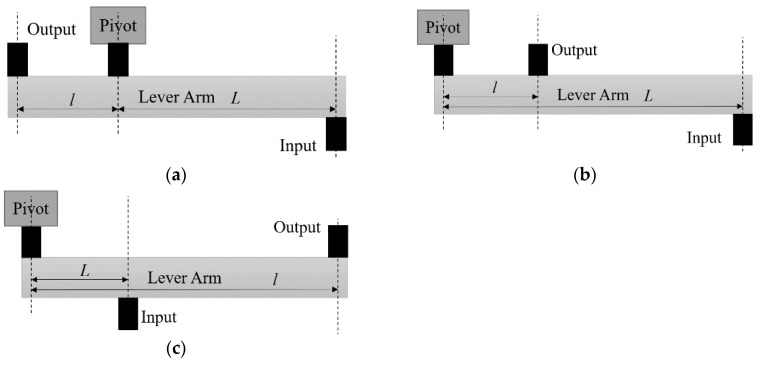
Classification of levers: (**a**) Type I; the pivot location is between input and output. (**b**) Type II; the output location is between pivot and input. (**c**) Type III; the input location is between pivot and output.

**Figure 2 micromachines-13-01201-f002:**
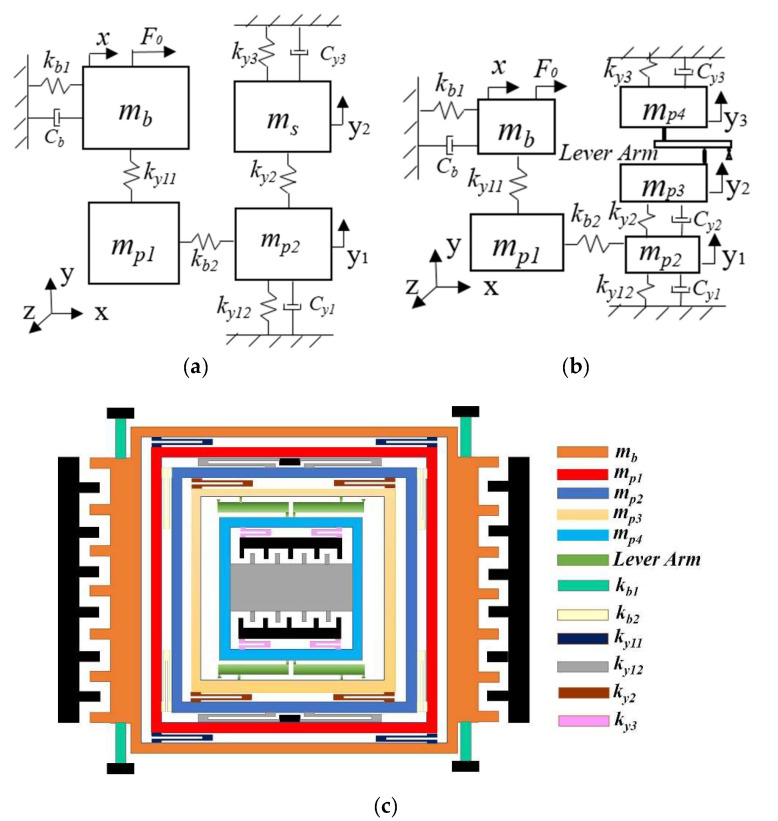
The 3-DOF micro-gyro with ALM: (**a**) sense system of micro-gyro without an ALM; (**b**) sense system of micro-gyro with an ALM; (**c**) physical schematic diagram of micro-gyro.

**Figure 3 micromachines-13-01201-f003:**
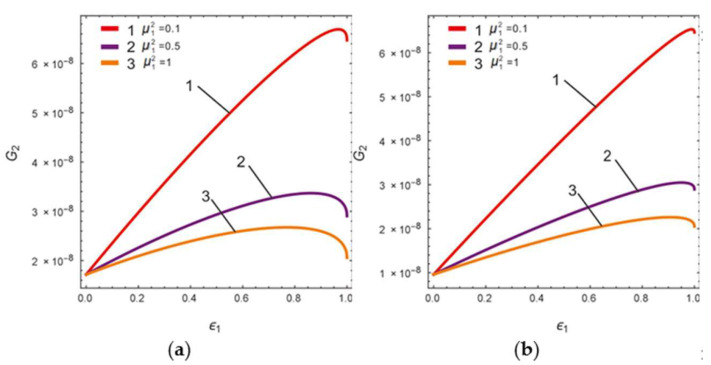
Gain G_2_ versus FCP $\epsilon_{1}$ curve: (**a**) *B* = 1, (**b**) *B* = 2.4.

**Figure 4 micromachines-13-01201-f004:**
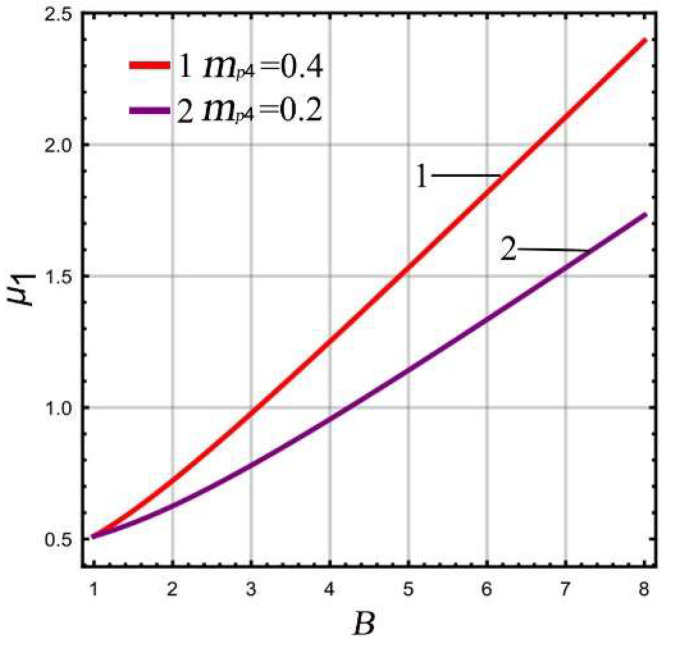
The relationship between $\mu_{1}$ and B.

**Figure 5 micromachines-13-01201-f005:**
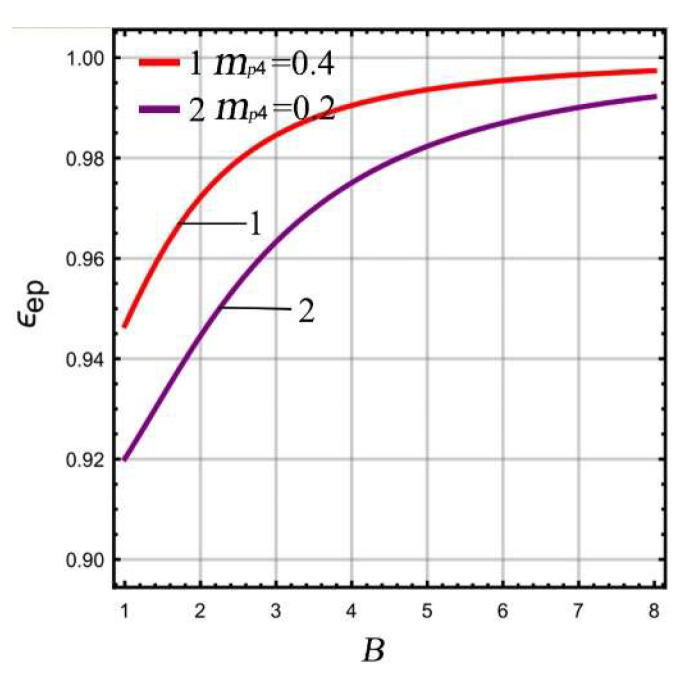
The relationship between $\epsilon_{ep}$ and B.

**Figure 6 micromachines-13-01201-f006:**
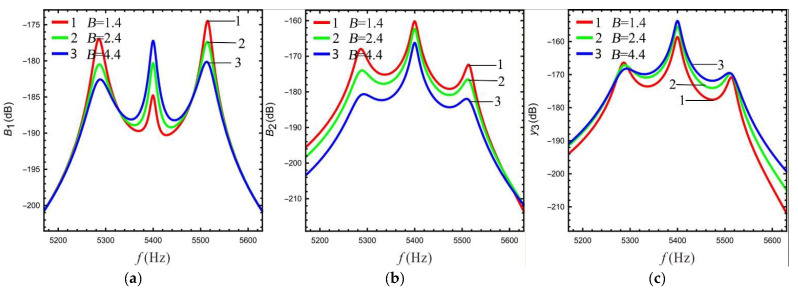
The relationship between gain and LR in sense mode: (**a**)sense-Ⅰ, (**b**) sense-Ⅱ (at lever input), (**c**) sense-Ⅱ (at lever output).

**Figure 7 micromachines-13-01201-f007:**
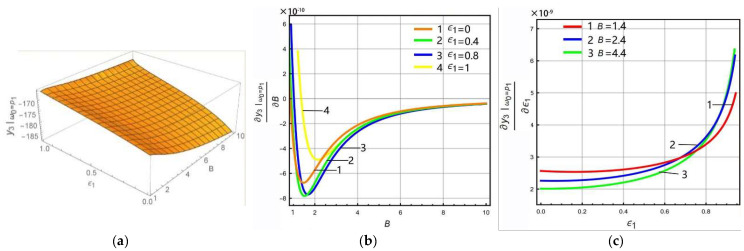
Variation rule of left peak ($\omega_{0}=p_{1}$): (**a**) 3D diagram, (**b**) gain derives the partial differential for LR, (**c**) gain derives the partial differential for FCP.

**Figure 8 micromachines-13-01201-f008:**
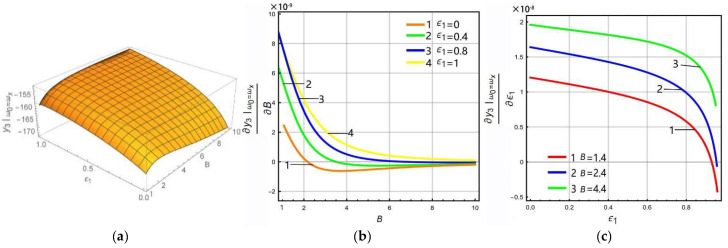
Variation rule of Coriolis peak ($\omega_{0}=\omega_{x}$): (**a**) 3D diagram, (**b**) gain derives the partial differential for LR, (**c**) gain derives the partial differential for FCP.

**Figure 9 micromachines-13-01201-f009:**
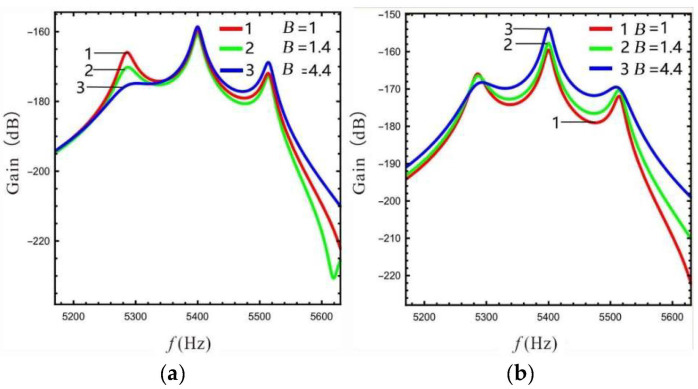
Effect of LR on the gain of the micro-gyro: (**a**) $\epsilon_{1}=0.4$, (**b**) $\epsilon_{1}=\epsilon_{ep}$ (B = 1.4, $\epsilon_{ep}=0.93$; *B* = 4.4, $\epsilon_{ep}=0.98$ ).

**Figure 10 micromachines-13-01201-f010:**
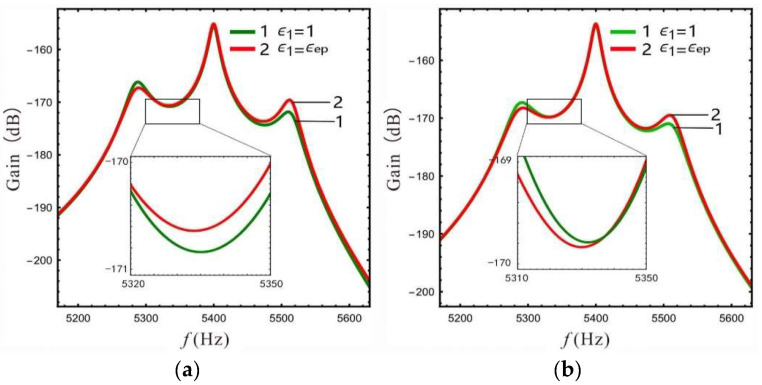
Gain is affected by FCP: (**a**) *B* = 2.4, (**b**) *B* = 4.4.

**Figure 11 micromachines-13-01201-f011:**
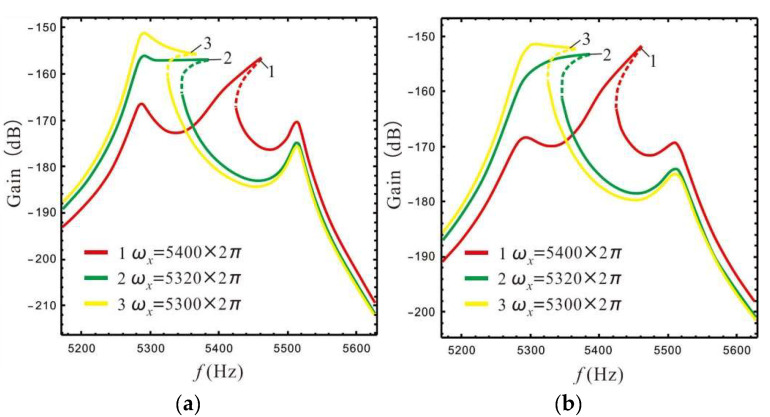
Gain and bandwidth affected by spacing between Coriolis and left peaks: (**a**) *B* = 1.4, (**b**) *B* = 4.4.

**Figure 12 micromachines-13-01201-f012:**
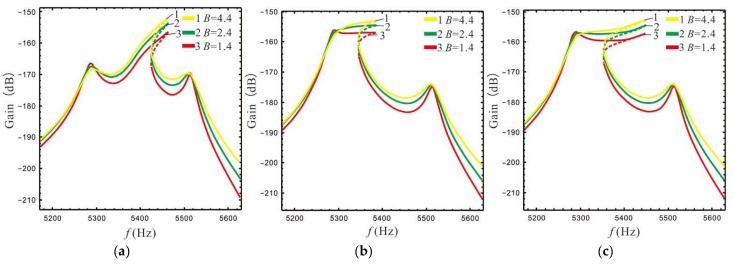
The shape of the frequency response curve is affected by LR: (**a**) $\omega_{x}=5400\times 2\pi k_{d}=12.2$, (**b**) $\omega_{x}=5320\times 2\pi k_{d}=12.2$, (**c**) $\omega_{x}=5320\times 2\pi k_{d}=12.5$.

**Figure 13 micromachines-13-01201-f013:**
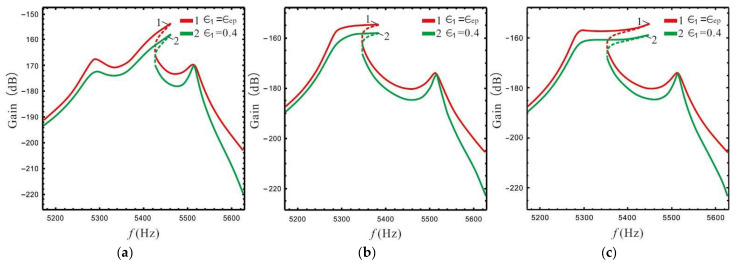
The shape of the frequency response curve is affected by FCP: (**a**) $\omega_{x}=5400\times 2\pi k_{d}=12.2$, (**b**) $\omega_{x}=5320\times 2\pi k_{d}=12.2$, (**c**) $\omega_{x}=5320\times 2\pi k_{d}=12.5$.

**Figure 14 micromachines-13-01201-f014:**
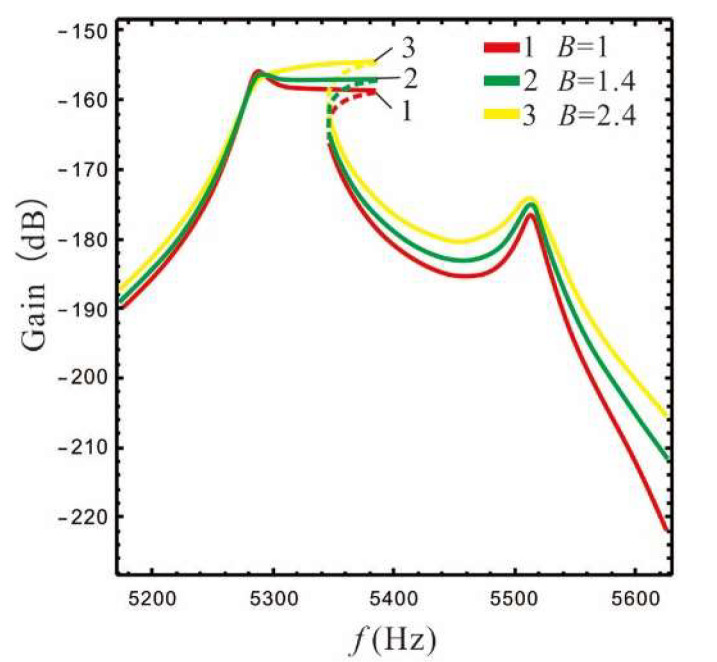
Gain is affected by the LR: $\epsilon_{1}=\epsilon_{ep}(B=1.4,\epsilon_{ep}=0.93;$$B=2.4,\epsilon_{ep}=0.95)$.

**Figure 15 micromachines-13-01201-f015:**
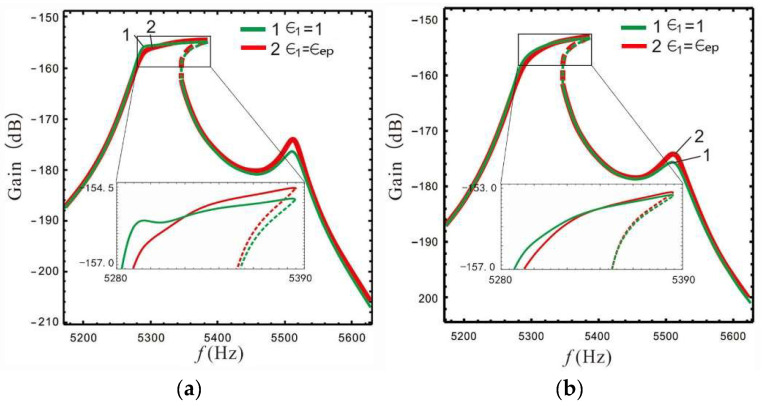
Gain is affected by FCP: (**a**) *B* = 2.4, (**b**) *B* = 4.4.

**Figure 16 micromachines-13-01201-f016:**
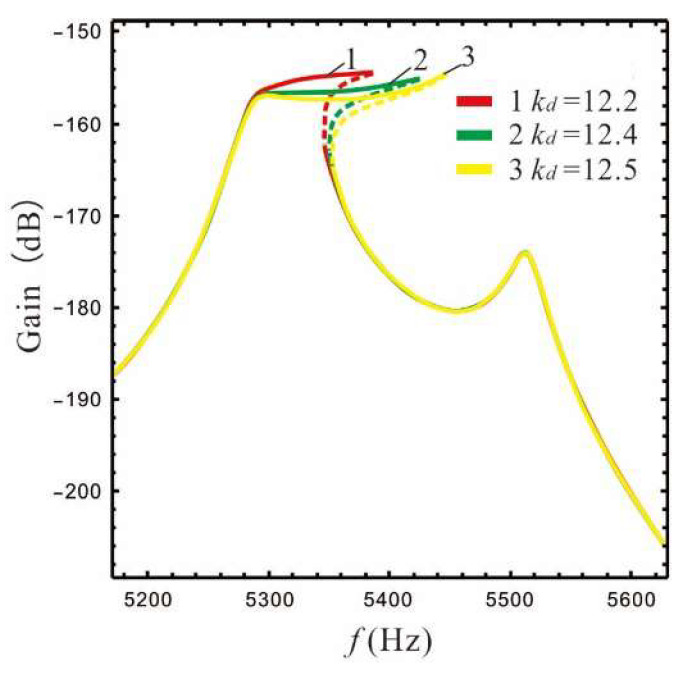
Gain and bandwidth are affected by nonlinear coefficients.

**Figure 17 micromachines-13-01201-f017:**
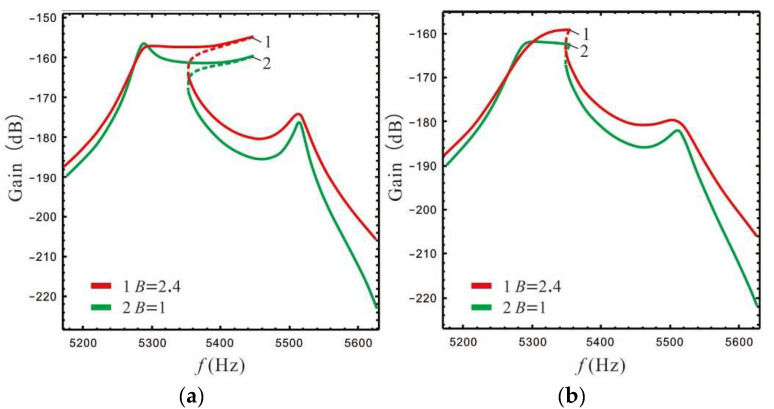
Gain and bandwidth are affected by damping coefficient for $k_{d}=12.5$; (**a**) $\omega_{x}=5320\times 2\pi ,\;P=10\;\mathrm{Pa}$, (**b**) $\omega_{x}=5320\times 2\pi ,P=20\;\mathrm{Pa}$.

**Table 1 micromachines-13-01201-t001:** The parameters of the micro-gyro.

Parameters	Values	Parameters	Values
*t*	80 µm	$l_{cap}$	16 × 10^−6^ µm
*m_b_*	2.85 × 10^−7^ Kg	$y_{comb}$	10 × 10^−6^ µm
*m* _*p*1_	2.6 × 10^−7^ Kg	$y_{cap}$	4 × 10^−6^ µm
*m* _*p*2_	2 × 10^−7^ Kg	$F_{0}$	5.34 × 10^−6^ N
*m* _*p*3_	1 × 10^−7^ Kg	$c_{b}$	4.5 × 10^−5^ N·s/m
*m* _*p*4_	0.2 × 10^−7^ Kg	$c_{y1}$	3.4 × 10^−5^ N·s/m
*N_comb_*	270	$c_{y2}$	7.088 × 10^−6^ N·s/m
*N_cap_*	500	$c_{y3}$	1.07 × 10^−5^ N·s/m
*l_comb_*	40 × 10^−6^ µm	$l_{cap}$	16 × 10^−6^ µm

**Table 2 micromachines-13-01201-t002:** Gain and bandwidth of linear micro-gyro under typical parameters.

Δ=230 $,\omega_{x}=5400\times 2\pi$	Gain (dB)	$y_{3}\mathrm{Growth}\;\mathrm{Rate}\;(\%)$	Bandwidth (Hz)	Bandwidth Growth Rate (%)
*B* = 1 (without an ALM)	−174.2	/	64	/
*B* = 1.4, $\epsilon_{1}$= 1	−173.2	12.5	67	4.7
*B* = 1.4, $\epsilon_{1}$ = $\epsilon_{ep}(0.93)$	−172.7	18.5	68	6.3
*B* = 2.4, $\epsilon_{1}$= 1	−170.8	47.1	73	14.1
*B* = 2.4, $\epsilon_{1}$ = $\epsilon_{ep}(0.95)$	−170.6	50.9	98	53.1
*B* = 4.4, $\epsilon_{1}$= 1	−169.7	66.0	94	46.9
*B* = 4.4, $\epsilon_{1}$ = $\epsilon_{ep}(0.98)$	−169.9	65.1	90	40.6

**Table 3 micromachines-13-01201-t003:** Gain and bandwidth of nonlinear micro-gyro under typical parameters.

Δ=230 $,\;\omega_{x}=5320\times 2\pi$	Gain (dB)	$y_{3}\;\mathrm{Growth}\;\mathrm{Rate}$(%)	Bandwidth (Hz)	Bandwidth Growth Rate (%)
B=1 (without an ALM)	−159.5	/	106	/
$B=1\;.4,\;\epsilon_{1}=1$	−158.7	9.6	108	1.9
$B=1\;.4,\epsilon_{1}=\epsilon_{ep}\left(0.93\right)$	−159.6	−1.1	109	2.8
$B=2.4,\epsilon_{1}=1$	−158.3	14.8	105	−0.9
$B=2.4,\epsilon_{1}=\epsilon_{ep}\left(0.95\right)$	−157.8	21.6	100	−5.7
$B=4.4,\epsilon_{1}=1$	−156.7	38.0	89	−16.0
$B=4.4,\epsilon_{1}=\epsilon_{ep}\left(0.98\right)$	−156.6	39.6	86	−18.9

## Data Availability

The data presented in this study are available on request from the corresponding author.

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
