# Peer review of "Dynamical Design and Gain Performance Analysis of a 3-DoF Micro-Gyro with an Anchored Leverage Mechanism"

_micromachines, 2022, doi:10.3390/mi13081201_

Round 1

Reviewer 1 Report

The authors proposed new Gyro based on Anchored Leverage Mechanism to improve its dynamic and overall performance of the device. The paper is well written and organised. The implementation and mathematical model are presented well. However, I have the following concerns need to be addressed before it gets published:

1- Is it piratical to design and fabricate a 3-DOF micro-gyro with an ALM? If yes, which MEMS fabrication process is suitable?

2- The model presented based on Newton’s second law is not new and I would like to see the coupling terms specifically the nonlinear stiffness components. These have significant contributions compared to the linear terms.

3- How sensitive is the device compared with the existing devices?

4- Which direction has a larger bandwidth and amplification? 

Other than these comments, the paper is deserved publication in Micromachines.

Reviewer 2 Report

Dynamical Design and Gain Performance Anaysis of 3-Dof Micro-gyro with an Anchored Leverage Mechanism - Review

References and literature review must be extensively revised and improved.

This paper describes a design for a non-resonant 3-DoF micro-gyro which employs the anchored lever mechanism to improve the gain and bandwidth of the sensor. The authors propose and analyze two different micro-gyros, namely linear and nonlinear ones. An analysis is performed to analyze the effect of variation of the leverage ratio and frequency coupling parameters, nonlinearity and damping coefficient on gain and bandwidth considering the anchored lever mechanism. Particularly, both the linear and nonlinear micro-gyro designs with the anchored lever mechanism are compared with the linear one without the anchored lever mechanism. The comparison is illustrated in detail, demonstrating the effective improvement of the sensor performance when the anchored lever mechanism is included in the design. However, the paper should be revised according to the following points:

1) In the introduction, the authors should insert some details about resonant micro-gyros, as they constitute the alternative proposed in the literature to the non-resonant structure, as the one developed by the author in this work. Indeed, non-resonant gyros have often complex mechanical structures as compared to resonant gyros. Moreover, electrostatic tuning can be exploited in the resonant gyros to avoid the mismatch between drive and sense mode frequencies and improve the performance of the sensor.

2) In Figure 2a and 2b, authors should indicate the displacements of masses ms,mp2,mp3,mp4 for the sake of clarity. In addition, it would be better to indicate the components of the mechanical structure represented in Figure 2c with a separated legend and add more details about the working principle of the proposed structure design.

3) In Figure 6, 7,8, 9, 10 I suggest to the author to add a legend to distinguish the different curves.

4) A list of the symbols used in the paper could be useful for the purpose of clarity.
